# Reversible switching of the environment-protected quantum spin Hall insulator bismuthene at the graphene/SiC interface

Niclas Tilgner [1,2], Susanne Wolff [1,2], Serguei Soubatch [3,4], Tien-Lin Lee [5], Andres David Peña Unigarro [1,2], Sibylle Gemming [1,2], F. Stefan Tautz [3,4,6], Thomas Seyller [1,2], Christian Kumpf [3,4,6] ✉, Fabian Göhler [1,2] ✉ & Philip Schädlich [1,2] ✉

Quantum spin Hall insulators have been extensively studied both theoretically and experimentally because they exhibit robust helical edge states driven by spin-orbit coupling and offer the potential for applications in spintronics through dissipationless spin transport. Here we show that a single layer of elemental Bi, formed by intercalation of an epitaxial graphene buffer layer on SiC(0001), is a promising candidate for a quantum spin Hall insulator. This layer can be reversibly switched between an electronically inactive precursor state and a bismuthene state, the latter exhibiting the predicted band structure of a true two-dimensional bismuthene layer. Switching is accomplished by hydrogenation (dehydrogenation) of the sample. A partial passivation (activation) of Si dangling bonds causes a lateral shift of Bi atoms involving a change of the adsorption site. In the bismuthene state, the Bi honeycomb layer is a prospective quantum spin Hall insulator, inherently protected by the graphene sheet above and the H-passivated substrate below.

Emerging quantum materials, distinguished by different types of strongly correlated electrons in solids, may pave the way towards next-generation applications such as dissipationless quantum transport or quantum computing[1]. In two-dimensional (2D) quantum spin Hall (QSH) materials, topologically protected spin-polarized edge states exist at zero magnetic field[2]. As noted by Kane and Mele[3,4], graphene represents such a material, however, with an extremely small topological band gap in the sub-meV range. In contrast, single honeycomb layers of heavy atoms, such as Bi[5,6], are considered to be ideal quantum spin Hall insulators (QSHI) due to the strong spin-orbit coupling that results in a substantial bulk band gap[7,8]. Such honeycomb layers have been realized experimentally, e.g., in the form of indenene, which can be considered as a hidden honeycomb despite its triangular structure[9],

and bismuthene[10] on SiC, where non-trivial gaps can open due to spin-orbit coupling and the orbital filtering effect[7,8,11], an effect that moves the $p_z$ orbital away from the Fermi level due to hybridization with the underlying substrate's dangling bonds. For bismuthene on SiC, its topologically non-trivial nature has been established in previous theoretical works by calculating the $\mathbb{Z}_2$ topological invariant, which distinguishes topological ($\mathbb{Z}_2 = 1$) from trivial ($\mathbb{Z}_2 = 0$) insulators. These studies found $\mathbb{Z}_2 = 1$, thereby confirming the QSH phase in this system[10,12,13]. In addition, the topological character of the system has been further confirmed by the presence of helical edge states crossing the bulk band gap in finite nanoribbons[7]. However, thin metal films are problematic for use in device applications due to their sensitivity to the environment from which they need to be protected.

[1]Institute of Physics, Chemnitz University of Technology, Chemnitz, Germany. [2]Center for Materials, Architectures and Integration of Nanomembranes (MAIN), Chemnitz, Germany. [3]Peter Grünberg Institut (PGI-3), Forschungszentrum Jülich, Jülich, Germany. [4]Jülich Aachen Research Alliance (JARA), Fundamentals of Future Information Technology, Jülich, Germany. [5]Diamond Light Source Ltd, Harwell Science and Innovation Campus, Didcot, Oxfordshire, UK. [6]Experimentalphysik IV A, RWTH Aachen University, Aachen, Germany. ✉e-mail: c.kumpf@fz-juelich.de; fabian.goehler@physik.tu-chemnitz.de; philip.schaedlich@physik.tu-chemnitz.de

The intercalation of epitaxial graphene on SiC is a well-suited way to produce protected metallic films. Known as confinement heteroepitaxy[14], this technique has emerged as a promising way to stabilize 2D quantum materials such as 2D gallium, indium, tin, and lead[14–17]. In this process, the intercalant moves to the interface, decouples the graphene buffer layer (zeroth-layer graphene, (ZLG) from the substrate and transforms it into a quasi-freestanding graphene (QFG) layer. The QFG layer subsequently protects the intercalated layer against environmental degradation[15]. Intercalation of Bi under the buffer layer has been the subject of previous studies, but the initial focus was primarily on its effect on the graphene properties[18–20]. For the Bi layer, however, a honeycomb lattice was predicted to be the energetically most favorable structure at a coverage of 2/3 of a monolayer[21]. And indeed, such a layer can be obtained experimentally by Bi deposition and subsequent annealing[19,20]. Here we demonstrate that this Bi layer is a precursor phase for 2D bismuthene, and that it can be reversibly transformed into bismuthene by hydrogenation. Moreover, we explain the mechanism driving the phase transition at the atomistic level and present the electronic and geometric structures of both phases. In the bismuthene state, the Bi layer exhibits all ingredients for a 2D QSH material.

## Results

### Electronic and geometric properties of the precursor phase

We start with a discussion of the properties of the precursor phase, which is prepared by Bi intercalation of a graphene buffer layer and several subsequent annealing steps (see Methods section for details). It has been designated as Bi $\beta$ phase in recent literature[19,20], since its potential for transformation into 2D bismuthene has not been fully

recognized before. Low-energy electron diffraction (LEED, see Fig. 1a) reveals a well-ordered layer system with $(\sqrt{3} \times \sqrt{3})R30°$-periodicity. The electronic structure as determined by angle-resolved photoelectron spectroscopy (ARPES) is shown in Fig. 1b. The $\pi$-bands of graphene are clearly observed with the Dirac cone at the $K_G$ point, indicating that the graphene is well decoupled from the Bi layer and the substrate below. Figure 1c shows a cut through the $K_G$ point in the plane perpendicular to that shown in Fig. 1b. The Bi layer, however, shows no signatures that would be characteristic of the 2D material bismuthene, but rather a weakly dispersing band -1.0–1.5 eV below the Fermi energy $E_F$, as indicated by the black line in Fig. 1b marking the maxima of this band. This will change once the transformation from precursor to bismuthene is performed, as discussed below.

To date, the structure of this phase has not been solved unambiguously, although the question of the Bi adsorption site in particular is a crucial one. It determines the bonding configuration between the Bi atoms and the substrate, and thus also within the Bi layer. We have addressed this question by applying an extension of the normal incidence X-ray standing wave (NIXSW) technique[22,23], known as NIXSW imaging[24]. It is based on a Fourier component analysis, which provides the element-specific density distribution of all atomic species in the unit cell, and since it is not susceptible to the phase problem of diffraction methods, it is able to resolve the full three-dimensional structure of the bulk and the surface. Details of the technique are given in the "Methods" section. A full discussion of the analysis of this data set is published elsewhere[25].

Figure 1d–f shows the results for the precursor phase. 2D cuts through the three-dimensional atomic density distribution for the bulk species Si and C are shown in 1d and 1e, respectively, and clearly

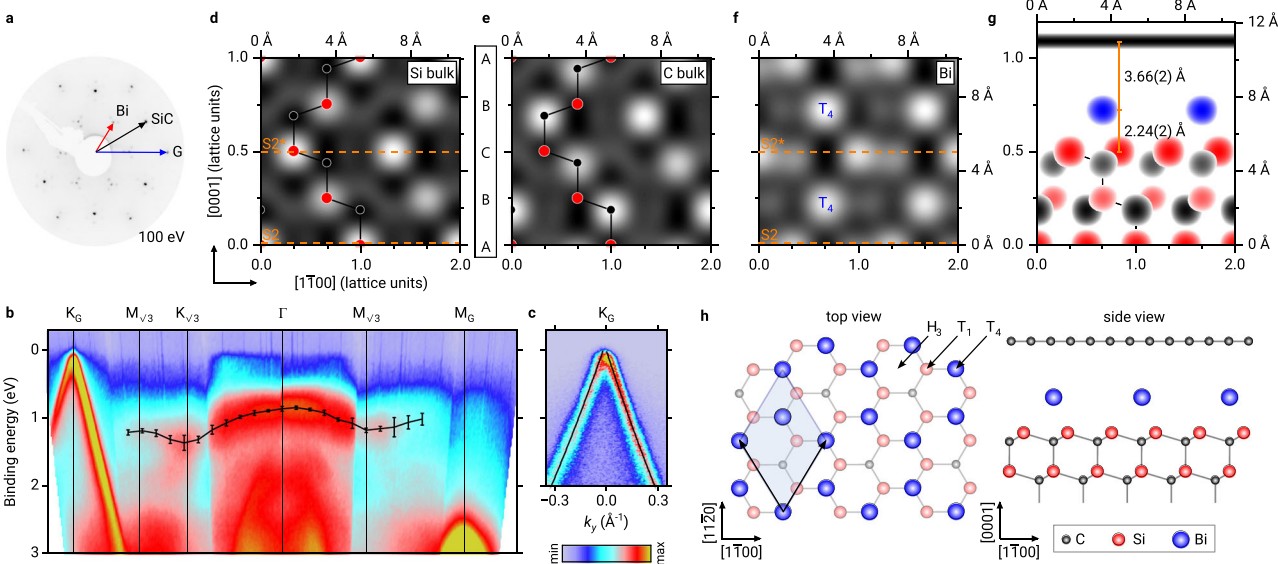

**Fig. 1 | Electronic and geometric structure of the precursor phase. a** LEED pattern with one spot of graphene, SiC, and Bi marked by a blue, black, and red arrow, respectively ($E = 100$ eV). **b** Energy-momentum cuts along the $\Gamma$-$K_G$ and $\Gamma$-$M_G$ directions of graphene. The weakly dispersing low-energy state originating from the precursor is marked by a black line with error bars derived from fitting the corresponding energy distribution curves. **c** Energy-momentum cut at $K_G$ in the direction perpendicular to the map shown in (**b**). The black line indicates the linear $\pi$-band dispersion, obtained from fitting the momentum distribution curves in the nearest-neighbor tight-binding approximation for graphene. **d–f** NIXSW imaging results for bulk-Si, bulk-C, and Bi, respectively. The Fourier-reconstructed electron densities in a plane spanned by the [1$\bar{1}$00] and [0001] directions are shown, white maxima correspond to the positions of the atoms. A ball-and-stick model of the bulk structure is superimposed on (**d, e**) and fits to the experimental finding very well. In **f** the clearest maxima are located at $T_4$ sites, indicating the adsorption sites

of Bi. Orange dashed lines in (**d, f**) indicate the SiC surface terminations S2 and S2* (A and C planes). **g** Complete structure of the graphene-protected precursor sample, as derived from NIXSW imaging for the S2* termination. All maxima from (**d** to **f**) that are higher than a certain threshold are reproduced (on a logarithmic color scale) and displayed in red, gray, and blue, indicating the positions of bulk-Si, bulk-C, and Bi, respectively. Note that in order to mimic the ball-and-stick model shown in (**h**), also maxima from a second adjacent plane, offset by half a lattice vector, are shown for the bulk species. Graphene, since it is incommensurate with the bulk structure, is shown as a horizontal black bar at its corresponding height. **h** Sketch of the atomic arrangement for the precursor structure on SiC(0001) in a $(\sqrt{3} \times \sqrt{3})R30°$ superstructure. Bi is located on the $T_4$ hollow sites, while the $H_3$ hollow and $T_1$ on-top sites are unoccupied. The graphene layer has been omitted in the top view for clarity.

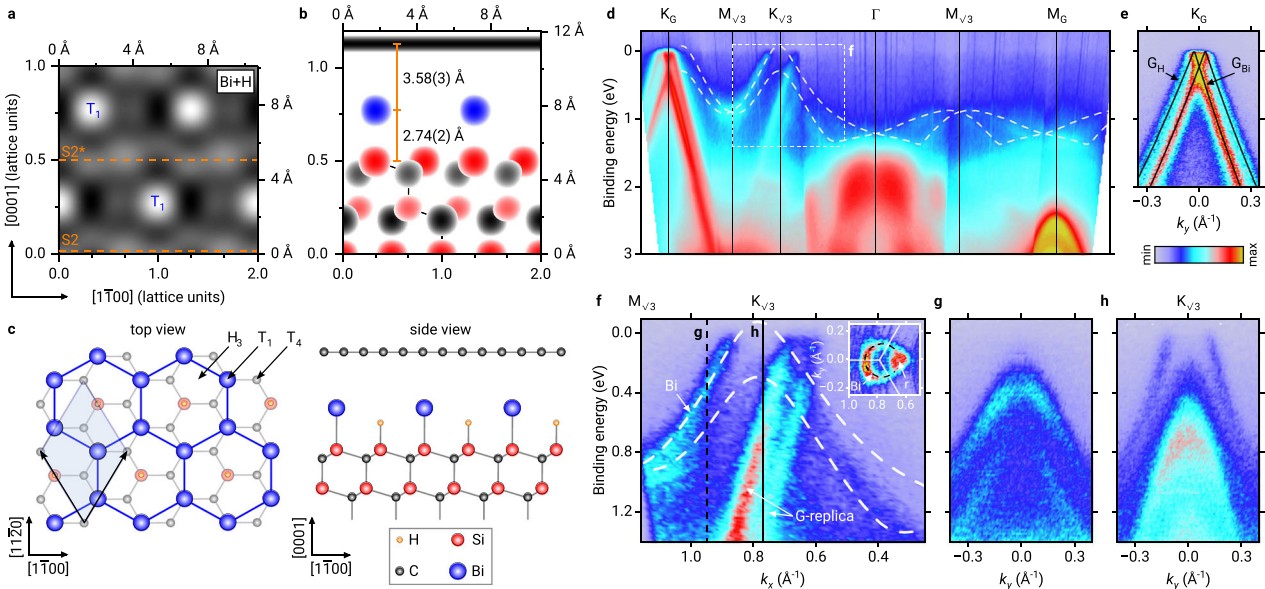

**Fig. 2 | Geometric and electronic structure of the 2D bismuthene layer on SiC(0001) (after hydrogenation). a** NIXSW imaging result for Bi, shown for the plane spanned by the [1$\bar{1}$00] and [0001] directions. White maxima correspond to the positions of the atoms. **b** Structure of the graphene-protected bismuthene sample, as derived from NIXSW imaging for the S2* termination. Red, gray, and blue circles represent the maxima of the Fourier-reconstructed electron densities for bulk-Si, bulk-C, and Bi, respectively. The incommensurate graphene layer is shown as a horizontal black bar at its corresponding height. **c** Sketch of the atomic arrangement of the bismuthene honeycomb on the SiC(0001) surface in a ($\sqrt{3} \times \sqrt{3}$)R30° superlattice. The graphene layer has been omitted in the top view for clarity. **d** Energy-momentum band maps along the $\Gamma$-$K_G$ and $\Gamma$-$M_G$ directions of graphene. The white dashed lines represent the valence band structure of bismuthene as calculated by DFT. **e** Energy-momentum map around $K_G$ of graphene in

the direction perpendicular to (**d**). Two energetically shifted Dirac cones of graphene are observed, due to the coexistence of graphene areas on bismuthene ($G_{Bi}$) and hydrogen intercalated graphene areas ($G_H$). Black lines were obtained by fitting the maxima of momentum distribution curves with the nearest-neighbor tight-binding band structure of graphene. **f** Close-up in the vicinity of $K_{\sqrt{3}}$ as marked in (**d**) by a white dashed rectangle. The inset shows the Fermi contour of the bismuthene in a $k_x$-$k_y$ map. Brillouin zone boundaries are drawn as white lines. The left part of the contour, indicated by a black dashed line, corresponds to the bismuthene contribution, in the right part it is superimposed by the graphene replica cones (marked r). **g**, **h** $E$-$k_y$ energy-momentum maps in the direction perpendicular to the map shown in (**f**), at the position indicated by a black vertical dashed line in (**f**) and at the $K_{\sqrt{3}}$ point of bismuthene, respectively.

resemble the well-known bulk crystal structure of 4H-SiC, as illustrated by the superimposed ball-and-stick model. However, the most relevant result, i.e., the density distribution for Bi shown in Fig. 1f, is more difficult to interpret, as it involves two different surface terminations: It is known that 4H-SiC with its atomic plane stacking sequence A-B-C-B-A undergoes step bunching when annealed for surface preparation, because the A- and C-terminated terraces are energetically favored over the B termination[26]. Atomic force microscopy results shown in Section 1 of the Supplementary Information indicate that this is also the case for our samples: We find a balanced distribution of the so-called S2 (A-plane) and S2* (C-plane) surface terminations. In Fig. 1d, these two terminations are indicated by horizontal, orange dashed lines, marking the positions of the uppermost Si atoms (surface planes). We also indicate these planes in Fig. 1f, because the Bi atomic positions, as obtained by NIXSW imaging, have to be evaluated relative to one of these planes. We find two maxima in the Bi density distribution (marked by $T_4$), each of which is located at a reasonable vertical distance to one of the surface planes, and therefore represents a Bi adsorption site. Their vertical distance to the surface plane is 2.24 Å, smaller than the expected length of a Si-Bi covalent bond (2.67 Å[27]), but consistent with a covalent bond between Si and a Bi atom on a hollow site. The lateral position of the maxima in Fig. 1f indicates that Bi is located above the C atom of the terminating SiC bilayer (compare with Fig. 1e), that is, the $T_4$ adsorption site. Note that the other (weaker) maxima in the map do not appear at reasonable distances from either of the two surface planes. The strongest of these, if interpreted as an adsorption site, would correspond to the $T_4$ site above a B-terminated surface, which may even be present on the surface to some small extent. However, it is more likely that the smaller maxima are artefacts due to the finite sum of Fourier components

(finite number of different Bragg reflections) used in the experiment (see Methods).

We conclude that in the precursor phase the Bi atoms adsorb exclusively on $T_4$ sites. Figure 1g shows the complete structure for the example of a S2* (C-plane) terminated substrate: All maxima (above a certain threshold) of the maps (1d–f) are shown in one plot, color-coded red, gray, and blue for Si, C, and Bi, respectively. The position of the graphene layer above Bi is also shown here, as a black bar 3.66 Å above the Bi layer. No lateral structure can be resolved for graphene since it is incommensurate with the bulk lattice. However, the vertical distance between Bi and graphene is close to the sum of the van der Waals radii of the two species involved (3.77 Å[28]), indicating that the Bi-graphene interaction is almost exclusively van der Waals-like. Note that our finding of Bi in $T_4$ hollow sites, although clearly evidenced by NIXSW imaging, is in contrast to the model discussed by Sohn et al.[19] who suggested the other hollow site of the SiC surface, $H_3$ (see Fig. 1h, top view). However, our finding is consistent with theoretical predictions for the most favorable adsorption sites on the Si-rich ($\sqrt{3} \times \sqrt{3}$) R30° reconstruction of SiC(0001)[29,30]. These authors identified $T_4$ as the most favorable site, owing to strain relaxation in the uppermost SiC bulk layer. The two C atoms below the Bi adatoms relax downwards, increasing their bond angles closer to the ideal tetrahedral angle of 109.5°, while the third C atom in the ($\sqrt{3} \times \sqrt{3}$)R30° supercell, which has no adatom above it, shifts upwards. This type of strain relief is not possible in the $H_3$ configuration, where the uppermost C bulk atoms are equivalent[29,30].

The adsorption of Bi on a hollow site has decisive consequences for its bonding configuration. As can be seen from the ball-and-stick model shown in Fig. 1h (top view), on the $T_4$ site each Bi atom has three equidistantly located Si neighbors, and the Bi layer therefore saturates

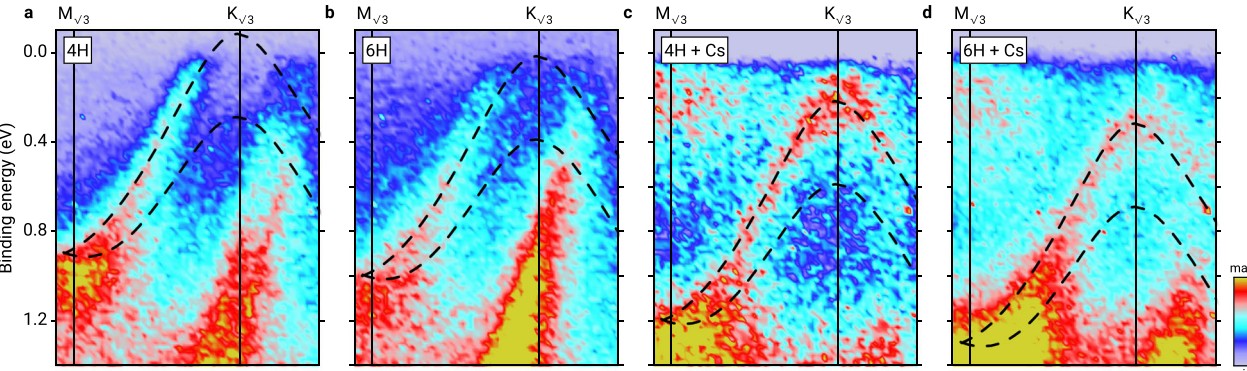

**Fig. 3 | Energy-momentum band maps of bismuthene at different doping levels.** Different doping levels were obtained by the use of different substrate polytypes 4H-SiC and 6H-SiC, which imply different spontaneous polarization strengths, and by additional adsorption of the alkali metal Cs on top of graphene, inducing a charge transfer into the bismuthene layer. **a** 4H-SiC substrate, no Cs. **b** 6H-SiC substrate, no Cs. **c** 4H-SiC substrate, with Cs. **d** 6H-SiC substrate, with Cs. In (**d**), the valence band maximum is positioned significantly below the Fermi level, indicating the presence of a band gap. Black dashed lines indicate the DFT calculated dispersion of the Bi Dirac cone. A rigid band shift was applied to the calculated bands to match the experimental results.

all Si dangling bonds. Note that we have drawn a Bi honeycomb here, since the results obtained for bismuthene (see below) are suggestive of this structure. However, since the localization of atoms by NIXSW imaging is restricted to the bulk unit cell, it is not possible to unambiguously discriminate between equivalent sites within the superstructure. In our case, the three $T_4$ sites within the $(\sqrt{3} \times \sqrt{3})$ unit cell cannot be distinguished. But adsorption sites other than $T_4$ can be excluded, and therefore each Bi atom forms bonds with three Si atoms, inhibiting a planar hybridization and thus the formation of a Bi layer of true 2D character. As we will demonstrate in the following, this is changed by a hydrogen-induced transition from the precursor phase to a true 2D bismuthene phase.

## Geometric structure of the 2D bismuthene phase

The transformation of the precursor phase into a 2D bismuthene phase is performed by hydrogenation in a two-step annealing process in a $H_2$ atmosphere (see Methods section for details). After hydrogenation, we have again performed NIXSW imaging, ARPES, and X-ray photoelectron spectroscopy (XPS) measurements. In Fig. 2a, b, we show the NIXSW imaging results for the sample after hydrogenation. While the results for the bulk species are unchanged (not shown), a comparison of Fig. 2a (after hydrogenation) with 1f (before hydrogenation) demonstrates that the position of the Bi atoms has changed, i.e., the atoms have rearranged throughout the Bi layer (see also Supplementary Information Section 2). Figure 2b shows the complete structure in the combined plot (similar to Fig. 1g), also illustrated by the corresponding ball-and-stick model in Fig. 2c (to be compared with Fig. 1h). These results reveal two important aspects: (i) Bi has moved away from the $T_4$ hollow site, which it occupied in the precursor phase. In bismuthene, it occupies the $T_1$ site on top of the uppermost Si atoms. (ii) The Bi layer did not decouple completely from the substrate by the hydrogenation process, but the Bi atoms are now located almost precisely at the distance of a covalent Si-Bi bond, at 2.74 Å above the topmost Si atoms. The sum of the covalent radii of the two species is 2.67 Å[27]. Thus, our structure determination indicates a vertically oriented, covalent bond between the Bi atoms on $T_1$ sites and the topmost Si atoms underneath, a result consistent with the structural model proposed by Reis et al. for bismuthene on a H-saturated SiC(0001) surface[10] (a system not covered by graphene). In addition, the Bi-graphene separation of 3.58 Å reveals a van der Waals like interaction between Bi and the overlying protective graphene layer.

The Bi adsorption site determines the bonding configuration also within the bismuthene layer. In fact, the change of the adsorption site from $T_4$ in the precursor phase to $T_1$ in bismuthene is the key for understanding the phase transition and the 2D properties of

bismuthene. In the precursor phase, each Bi atom saturates three substrate dangling bonds in the unit cell, see above. But the existence of three bonds to underlying Si atoms does not allow for the planar hybridization of Bi that is needed for the formation of the characteristic band structure of a 2D bismuthene honeycomb. After hydrogenation, when Bi adsorbs on $T_1$, only one covalent Bi-Si bond is formed, involving the $p_z$ orbital and one of the five valence electrons of Bi. The remaining $s$ and $p$ orbitals with four valence electrons allow for a planar hybridization within the Bi layer. Thus, this orbital filtering effect removes exactly one orbital (and valence electron) from the in-plane binding configuration. This is required to establish the formation of the distinct Dirac-like bands being characteristic for 2D bismuthene, but only possible when Bi is adsorbed on a $T_1$ on-top position above Si. As our experimental results suggest, the driving force behind this change of the adsorption site is the hydrogenation. The affinity to saturate Si dangling bonds appears to be higher for H than for Bi, but not to the extent that Bi can be completely expelled from the interface. Bi is apparently repelled from its triply coordinated $T_4$ sites, stabilized instead at the $T_1$ site by the single bond to the Si surface atoms. The formation of Bi-Bi bonds in the Bi honeycomb structure of bismuthene may also contribute to this stabilization.

## Electronic properties of 2D bismuthene underneath epitaxial graphene

The proof that the hydrogenated Bi layer is indeed a true 2D bismuthene layer comes from the determination of the electronic structure by means of ARPES. The results of these measurements are shown in Fig. 2d–h. From the $E$-$k_x$ energy-momentum band map shown in Fig. 2d it is evident that the weakly dispersing band discussed above for the non-hydrogenated precursor phase has vanished, and instead, strongly dispersing Dirac-like bands emerge close to $E_F$ between the $M_{\sqrt{3}}$ and the $K_{\sqrt{3}}$ points. In the close-up shown in Fig. 2f, these bands are even more clearly visible. The white dashed lines represent density functional theory (DFT) calculations for the QSHI state of bismuthene (see Supplementary Information, Section 3 for details). The steep bands next to them, labeled G-replica and crossing the Fermi level just to the right of the $K_{\sqrt{3}}$ point, are the replica of the graphene Dirac cones, backfolded due to the SiC lattice[31] and partly covering the bismuthene bands. Note that the replicas are not observed for the precursor phase (Fig. 1b) since the $(\sqrt{3} \times \sqrt{3})$ R30° superstructure of the $\beta$ phase requires a second-order process to produce scattering in this range of the $k$-space, making its intensity very small. On the bismuthene sample, however, some parasitic hydrogen-intercalated QFG domains exist (see Discussion below), which are $(1 \times 1)$ reconstructed and produce intensity of the G-replica in a first-order scattering

process. The upper one of the bismuthene calculated bands (upper white-dashed line) matches very well with the left of the Dirac-like bands visible in the ARPES data (marked Bi in Fig. 2f), while the lower band is better seen in an $E$-$k_y$-map perpendicular to the maps shown in Fig. 2d, f. Figure 2g shows such a perpendicular map through the position marked by the black dashed line labeled g in Fig. 2f. The splitting of these upper and lower Dirac-like bands, best to be seen in Fig. 2g, is a Rashba-type spin splitting, as demonstrated by the DFT calculations detailed in Section 3 of the Supplementary Information, and is thus ascribed to the inversion symmetry breaking at the interface. We also mention that the peak shapes of XPS measurements confirm that the bismuthene phase is metallic, in contrast to the precursor phase, which is insulating (see Supplementary Information, Section 4 for details).

Another fingerprint of a QSHI state is a band gap in the Dirac cone at the Fermi edge, i.e., a deviation from the linear dispersion expected for honeycomb lattices with negligible spin-orbit coupling. Such a gap is not directly visible in the data shown in Fig. 2, which is due to a significant p-doping of our bismuthene layer that shifts the bands upwards (in contrast to ref. 10). It is known from the literature that the p-doping is due to the pyroelectric nature of the substrate, where a residual bulk dipole moment persists, the magnitude of which depends on the SiC polytype[32]. At the surface, this spontaneous polarization acts as a pseudo-acceptor layer, generating a sheet of negative charge density that induces holes in adjacent materials[33,34]. For our case, the p-doping can be quantified as $p_{Bi} = (1.84 \pm 0.12) \times 10^{13}$ cm$^{-2}$. The p-doping of the bismuthene layer is well visible in the inset of Fig. 2f, which shows a $k_x$-$k_y$ section around $K_{\sqrt{3}}$ directly at the Fermi surface. The circular contour on the left side (black dashed line) is a horizontal cut through a Dirac cone which is still of finite size at the Fermi level. On the right, the ring is superimposed by a signal from the graphene replica cone (labeled $r$).

While this p-doping makes it impossible to observe the gap in the Dirac cone directly, it can be made visible in ARPES measurements performed at different doping levels, as presented in Fig. 3. These are energy-momentum maps along the $K_{\sqrt{3}} - M_{\sqrt{3}}$ direction, similar to Fig. 2f, but for different substrates and with Cs adsorbed additionally on top of the graphene. Both modifications cause downshifts of the band structure of different magnitude, due to (i) the 6H polytype exhibiting a weaker spontaneous polarization[34], and hence reducing the charge carrier concentration to $p_{Bi} = (4.1 \pm 1.5) \times 10^{12}$ cm$^{-2}$, and (ii) due to the alkali atoms providing some additional n-doping of the bismuthene without intercalating the Bi or graphene layer[35]. The downshift of these two measures is ≈0.1 eV due to the 6H substrate, and ≈0.3 eV due to Cs adsorption. The series of energy-momentum maps in Fig. 3a–d, measured for different polytypes and with or without Cs adsorption, shows a stepwise downshift of the bismuthene band structure. While in Fig. 3a, b (4H-SiC and 6H-SiC, respectively, no Cs), the valence band maximum is still above or near the Fermi level, in Fig. 3c, d (both substrates with Cs), it moves further down, clearly demonstrating the existence of a band gap.

Finally, it should be mentioned that the ARPES data shown in Fig. 2 also contain the band structure of graphene. Two sets of $\pi$-bands can be observed at the $K_G$ point, shifted in energy with respect to each other, as apparent in the $E$-$k_y$-map shown in Fig. 2e. One comes from the graphene layer on bismuthene, the other from areas that are not Bi-intercalated. Such areas arise during the preparation of the precursor phase. Before hydrogenation, in these areas, only the graphene buffer layer (ZLG) is present, which in the hydrogenation process turns into H-intercalated QFG. Since QFG and bismuthene areas have different doping levels, two Dirac cones arise. Before hydrogenation, we observe only one single Dirac cone (from the graphene on top of the precursor phase, see Fig. 1c), with the Dirac point located only slightly above the Fermi energy, indicating that this graphene layer is almost undoped.

We conclude from the experimentally obtained band structure of the hydrogenated Bi layer and its very good agreement with the DFT calculations−in particular from the existence of the Dirac-like band, its spin splitting, and the energy gap at the Fermi edge−that this Bi layer is indeed a single layer of 2D bismuthene. We thus have grown a bismuthene layer underneath graphene, protected from the environment, and hence solved a longstanding challenge.

**Reversibility of the precursor to bismuthene phase transition**

It is known that a purely H-intercalated QFG layer is only stable up to 700 °C in ultra-high vacuum (UHV)[36]. The formation of our bismuthene phase was carried out well below this temperature, by annealing at 550 °C in H$_2$ atmosphere (see Methods). Thus, the bismuthene is expected to be unstable above 700 °C. In Section 5 of the Supplementary Information, in particular in Fig. S6e, we show that indeed the transition from the precursor phase to the bismuthene phase is reversed by annealing at 700 °C in UHV. The electronic bands associated with bismuthene disappear and the weakly dispersing band characteristic of the precursor phase is reestablished. It is remarkable that both the precursor-to-bismuthene transformation by hydrogenation, as well as the bismuthene-to-precursor transformation by annealing (and dehydrogenation), are reversible. We have performed the hydrogenation−dehydrogenation cycle several times on one sample, and always found largely the same band structures in ARPES for both precursor and bismuthene phases. Thus, the 2D properties of bismuthene can be switched on and off by hydrogenation and dehydrogenation.

A common feature of both phases is the $(\sqrt{3} \times \sqrt{3})$R30° LEED pattern. This reconstruction, which indicates a Bi coverage significantly below 1 ML, appears to be an important ingredient of the precursor phase as well. This is also reflected by the fact that other intercalated Bi phases, e.g., the $\alpha$ phase consisting of a close-packed unreconstructed layer with (1 × 1) periodicity, cannot be converted into bismuthene, as discussed in Section 6 of the Supplementary Information. This can be understood by the mechanism behind the phase transition: the H-saturation of 1/3 of the Si dangling bonds, which in turn triggers the change of the Bi adsorption site from T$_4$ to T$_1$, is only possible when the Bi coverage is (locally) at or below 2/3 of a monolayer. At higher coverages, e.g., in the case of the $\alpha$ phase, all Si dangling bonds are saturated one by one by Bi atoms[20], which inhibits the H-saturation and thus the formation of the honeycomb structure of bismuthene.

## Discussion

In this study, we report the formation of a 2D bismuthene honeycomb layer, a prospective QSHI, sandwiched between a graphene layer above and the SiC substrate below. The evidence for having a true 2D bismuthene layer is the excellent agreement of our ARPES data with DFT results for the QSHI state of bismuthene (own calculations and refs. 7,10,21). In particular, the archetypal Dirac-like bands near $E_F$, a band gap around the $K_{\sqrt{3}}$ point, and a pronounced Rashba-type energy splitting of the bismuthene valence bands are clearly revealed in ARPES.

The bismuthene is formed in a reversible transition from a precursor, the $(\sqrt{3} \times \sqrt{3})$ R30° reconstructed intercalated Bi $\beta$ phase. The phase transition is driven by hydrogenation and involves a lateral movement of the Bi atoms from the T$_4$ hollow adsorption site to the T$_1$ site on top of the uppermost Si bulk atoms. This change in adsorption site is the key to understanding the phase transition, as it determines the bonding configuration in the Bi layer: in the precursor phase, each Bi atom has three equidistant Si neighbors, in the bismuthene phase, only one. Only in the latter case, the distinct Dirac-like bands form, since exactly one orbital (the $p_z$ orbital) is excluded from the in-plane hybridization. This orbital filtering is indispensable for the formation of the characteristic band structure of the 2D bismuthene honeycomb.

Stability is obviously a crucial aspect for any epitaxial 2D (multi-) layer system, in particular for future applications. Due to the encapsulation by the graphene layer above and the SiC substrate below, the bismuthene layer proved to be stable in air. In Section 5 of the Supplementary Information we demonstrate that the band structure of the graphene/bismuthene/SiC system has not changed after 24 h of exposure to air. The bismuthene is indeed efficiently protected from the environment. Thus, we have successfully fabricated a switchable, air-stable bismuthene QSHI with a large band gap and strong Rashba spin splitting. These results represent an important step forward on the road towards future QSH devices.

**Note added in proof:** During the review process, we became aware of a preprint reporting bismuthene intercalation results that are consistent with ours[37].

## Methods

### Sample preparation

Epitaxial ZLG substrates were prepared using the polymer-assisted sublimation growth on SiC[38], a method that is well-known to provide graphene samples with superior quality compared to those grown in UHV. SiC wafers were purchased from Pam-Xiamen. Bi intercalation was carried out following a dedicated deposition and annealing approach[19,20]: after degassing the samples in UHV at 450 °C, Bi was deposited on the SiC surface from a custom-built Knudsen cell (operated at a temperature of 550 °C) for 120 min to evaporate a thin layer of elemental Bi. Deposition took place in a dedicated chamber with a base pressure better than $5 \times 10^{-9}$ mbar. After in-vacuo transfer to the UHV analysis system, the samples were annealed at 450 °C for 30 min, which led to Bi intercalation of the ZLG and the formation of the $\alpha$ phase, a densely packed Bi layer underneath graphene.

Subsequently the precursor phase (Bi $\beta$ phase) was produced by annealing the $\alpha$ phase sample at 950 °C, leading to a partial depletion of Bi from the intercalation layer and the formation of a $(\sqrt{3} \times \sqrt{3})$ R30° superlattice, as verified by LEED (see Fig. 1a)[19,20]. The sample temperature during annealing was controlled by a pyrometer, assuming a sample emissivity of 0.9. After a fast transport through air (no longer than 5 min), the hydrogenation of the samples was carried out by exposure to ultra-pure $H_2$ (880 mbar, 0.9 slm) in a dedicated tube furnace[39]. Samples were first heated to 300 °C for 10 min, followed by the main hydrogenation process at 550 °C for 90 min. This led to the formation of the bismuthene phase in all regions of the sample surface that were Bi intercalated. Note that due to an overall lack of Bi, in some cases, small ZLG domains were formed in the precursor phase, which transformed into H-intercalated QFG upon hydrogenation. Judging from the primary experimental methods employed, which are ARPES and NIXSW, we can state that the samples were uniform in the sense that the ratio of the areas covered by bismuthene and H-intercalated QFG domains was constant across the entire sample. The footprint of the incident beams of both methods was in the range of a few 100 μm.

### Photoelectron spectroscopy & low-energy electron diffraction

After growth, the samples were transferred in vacuum into a UHV chamber dedicated for XPS, ARPES, and LEED. The chamber was equipped with a monochromatized SPECS Focus 500 Al-Kα X-ray source, a monochromatized SPECS UVS 300 ultraviolet light source providing linear polarized He-I and He-II radiation, and a SPECS Phoibos 150 hemispherical electron analyzer with the 2D-CCD detector. For LEED, a SPECS ErLEED 150 was used. The base pressure of the UHV system was better than $3 \times 10^{-10}$ mbar.

### Normal incidence X-ray standing wave (NIXSW) and NIXSW imaging

For the NIXSW experiments, the samples were brought to beamline I09 of the Diamond Light Source in Didcot, UK. A dedicated UHV transport chamber (vacuum suitcase) was used for the transport, which allowed transfer in UHV both in Chemnitz, where the samples were grown and pre-characterized, and at the Diamond beamline. The chamber was operated at a pressure better than $5 \times 10^{-9}$ mbar. The data were measured in the UHV system of the NIXSW endstation of beamline I09 using a Scienta EW4000 HAXPES analyzer. In such an NIXSW experiment, in general, XPS data is recorded from the sample surface while an X-ray standing wave field, generated by the interference of an incident and a Bragg reflected X-ray wave, is established in the bulk crystal and at its surface[22,23,40,41]. Given the well-defined position of the standing wave with respect to the crystal lattice, the photoelectron yield of all atomic species depends on the positions of the atoms in the unit cell. The method is therefore capable of localizing any ensemble of absorber atoms with respect to the Bragg planes of the reflection used to generate the standing wave field. It is most commonly used to measure the vertical position of adsorbates on surfaces or interlayer distances above crystalline substrates (see e.g., refs. 17,20,42–46, and references in refs. 22,23). For this standard NIXSW experiment, a Bragg reflection with a scattering vector perpendicular to the surface must be chosen. The technique provides separate structural information for all those species in the sample, the photoemission signals of which can be separated.

The NIXSW-based Fourier imaging technique (NIXSW imaging), which we applied in this work, goes a decisive step further: it exploits the fact that the result of one NIXSW measurement with any chosen Bragg reflection **H** = $(hkl)$ represents the **H**th Fourier component of the atomic density of the atomic species under consideration. Thus, collecting data using a sufficient number of different non-equivalent reflections allows the reconstruction of the element-specific atomic density relative to the bulk unit cell. In this way, the geometric structure of the sample (both bulk and surface) can be derived using the well-established formalism of inverse Fourier transforms. Therefore, this technique is not susceptible to the general phase retrieval problem inherent in diffraction techniques, as the NIXSW method provides both amplitude and phase information. Although this technique was first proposed about two decades ago[24], it has not been widely used until now due to its significant technical requirements. We are also not aware of any 2D material systems that have been studied using NIXSW imaging so far.

Here, we performed NIXSW measurements on seven inequivalent Bragg reflections using the core levels C 1s, Si 2s, and Bi $4f_{7/2}$. In the C 1s spectra, the signal from the graphene and bulk carbon can be separated easily due to a large core level shift. More details on the analysis of the NIXSW data sets can be found in ref. 25.

### Computational details

DFT, as implemented in the ABINIT package[47], was used to calculate the structural and electronic properties of the systems. For this purpose, fully relativistic norm-conserving pseudopotentials with the generalized gradient approximation in the Perdew-Burke-Ernzerhof[48] formulation were used to evaluate the exchange-correlation potential. Optimization of the initial structures were performed using a $k$-point grid of $8 \times 8 \times 1$ until residual forces smaller than $5 \times 10^{-6}$ eV/Å were achieved. For an accurate ground state description, a $12 \times 12 \times 1$ grid was adopted. In all cases, a 1088 eV cutoff energy was used for the plane wave basis set and spin-orbit coupling was considered. Spin-textures were plotted using the PyProcar[49] package. In our calculations, the graphene layer was not included in order to reduce the computational cost. Hence, we focus on the bismuthene phase on SiC. The dangling bonds at the lower carbon surface of SiC were passivated with H atoms and a vacuum layer of 15 Å was used.

## Data availability

The data generated in this study are available at Jülich DATA, the institutional research data repository at Forschungszentrum Jülich[50].

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

## Acknowledgements
The authors would like to thank Christoph Lohse for substrate preparations. We also thank the Diamond Light Source for access to beamline I09 through proposal SI36085-2 and the I09 beamline staff (Pardeep Kumar Thakur and Dave McCue) for technical support. This work was supported by the German Research Foundation (Deutsche Forschungsgemeinschaft, DFG) within the Research Unit FOR5242 (project 449119662) and the Collaborative Research Centre No. 223848855-SFB 1083, sub-project A12.

## Author contributions
N.T., T.S., C.K. and P.S. conceived the project. The sample preparation was carried out by N.T. The PES and LEED measurements at TU Chemnitz were performed and analyzed by N.T. and P.S. The NIXSW experiments were performed by N.T., S.W., S.S., T.-L.L., and C.K. and analyzed by N.T. and S.W., with significant input from S.S. and C.K. DFT calculations were performed and interpreted by A.D.P.U. and S.G. The atomistic model explaining the reversible phase transition was developed by N.T., F.S.T., T.S., C.K. and P.S. All authors discussed the results. N.T. made the figures and N.T., C.K., F.G. and P.S. wrote the paper, with significant input from all authors.

## Funding

## Competing interests
The authors declare no competing interests.
