## [Transparent Peer Review file · Nature Communications]

Reversible Switching of the Environment-Protected Quantum Spin Hall Insulator Bismuthene at the Graphene/SiC Interface

Corresponding Author: Professor Christian Kumpf

Version 0:

Reviewer comments:

Reviewer #1

(Remarks to the Author)

Monolayer honeycomb lattice systems with strong spin-orbit coupling (SOC) are long-sought candidates for a stable quantum spin Hall (QSH) insulator due to their potential to host a large SOC gap. However, realizing such samples requires sophisticated monolayer fabrication techniques such as epitaxial growth and intercalation. Tilgner et al. studied a 2D bismuth layer embedded in a SiC substrate and monolayer graphene using a recently developed technique (refs. 12,13). The authors used normal incidence x-ray standing wave (NIXSW) techniques and angle-resolved photoemission spectroscopy ARPES, aided by density functional theory (DFT) calculations to characterize the structural and electronic properties. While the validity of the NIXSW data analysis is difficult to judge since the authors state that the raw data and analysis process are published elsewhere, it seems that the overall data appear to be systematically analyzed and of high quality. In particular, the NIXSW results, which provide quantitative atomic positions near the surface, are impressive. Before recommending the manuscript for publication in Nature Communications, I have several technical questions and suggestions for improvement in terms of transparency, clarity, and impact of the study.

- 1) Given the general policy on data transparency, the minimal dataset necessary to support the conclusions (Fig. 1g and 2b) should be provided.
- 2) What do the ball sizes mean in Fig. 1g and 2b? Why does the adjacent plane appear smaller?
- 3) Can the graphene capping layer be exfoliated? This is crucial for the potential application of the QSH device.
- 4) The successful X-ray standing wave measurements suggest high sample uniformity, but can the authors quantify the uniformity of the bismuthene layer (e.g., uniform region size, concentration of disorder)
- 5) The reason why the backfolded graphene replica does not appear in the precursor phase should be discussed. I would suspect that, in bismuthene, Bi is partially depleted after hydrogenation, leading to stronger interaction between graphene and the SiC surface.

Reviewer #2

(Remarks to the Author)

In this work, N. Tilgner et al. report on the formation of a 2D bismuthene honeycomb layer at the graphene/SiC interface. The 2D bismuthene phase is transformed from a precursor phase by hydrogenation, while the precursor phase is obtained by Bi intercalation. Combining the ARPES results and DFT calculations, the 2D bismuthene is regarded as a promising candidate for a quantum spin Hall insulator. The 2D bismuthene phase is stable in air due to the protection of the graphene sheet above and the substrate below, and it can be switched to the precursor phase by de-hydrogenation. The findings in this work are practically interesting. Before the manuscript can be considered for publication in Nature Communications, several issues require clarification.

1. The NIXSW imaging results show that the Bi atoms in the bismuthene precursor phase are exclusively located on the T4 sites, while in the bismuthene phase, the Bi atoms occupy the T1 sites. Why do Bi atoms move from T4 sites to T1 sites after hydrogenation?
2. The ARPES results in the main text exhibit the band structures of bismuthene precursor and bismuthene. And ARPES data of the Bi α phase before and after hydrogen treatment are shown in the supplementary material. For comparison, what is the band structure of graphene/SiC system before Bi intercalation?

3. What is NIXSW imaging result of the bismuthene on 6H-SiC substrate?

Reviewer #3

(Remarks to the Author)

The work discusses the formation of a two-dimensional bismuthene honeycomb layer, a prospective quantum spin Hall insulator, sandwiched between a graphene layer above and the SiC substrate below.

The generation of quantum spin Hall insulator is certainly an important issue for the for applications of spintronics. The authors claim that a single layer of elemental Bi, formed by intercalation of an epitaxial graphene buffer layer on SiC(0001), can be reversibly switched between an electronically inactive precursor state and a bismuthene state. In the latter, the Bi honeycomb layer is a prospective quantum spin Hall insulator. However, I have three concerns and wish they could be clarified.

1. Validation of Topological Protection

The topologically protected properties are only indirectly verified via ARPES, lacking direct evidence from quantized conductance or non-local transport experiments. It is recommended to supplement the study with transport measurements on nanoribbon devices or theoretical calculations of edge-state bands in nanoribbons to confirm the quantum spin Hall effect.

2. Reversible Switching Stability

While the reversible switching between the precursor and bismuthene states is mentioned, the cycling stability (e.g., structural/performance repeatability after multiple hydrogenation/dehydrogenation cycles) remains unaddressed. Additional data demonstrating the robustness of switching over multiple cycles would strengthen the claims.

3. Impact of Graphene Protection on Transport

Although the bismuthene layer exhibits a wide topological gap under strong spin-orbit coupling, the metallic Dirac cones in the graphene overlayer might interfere with electronic transport. The authors should clarify whether the quantum spin Hall edge states would experience scattering into the graphene layer, potentially degrading the dissipationless spin transport properties.

Reviewer #4

(Remarks to the Author)

The manuscript presents a study on the reversible switching of a bismuthene-based quantum spin Hall insulator intercalated between graphene and SiC(0001). While the work demonstrates competent materials engineering and thorough characterization, the core findings represent an incremental advance over prior studies (e.g., Reis Science 2017, Schmitt Nat. Commun. 2024), although the material is different. The work would be more appropriately suited for specialized materials journals, where detailed studies of material stabilization and property modulation are regularly featured.

So, I do not recommend this article to be published in Nature Communications.

Version 1:

Reviewer comments:

Reviewer #1

(Remarks to the Author)

The authors have carefully addressed the questions and comments. I believe the manuscript is now suitable for publication in Nature Communications.

Reviewer #2

(Remarks to the Author)

Authors have addressed my concerns. I recommend the publication of this work.

Reviewer #3

(Remarks to the Author)

The authors have responded to my comments satisfactorily. Therefore, I recommend a publication of this revised manuscript in Nature Communications, and I believe this work will be of great interest to the community of two-dimensional materials.

Reviewer #4

(Remarks to the Author)

I disagree with the authors' claim that this work is "an significant step ahead towards a QSHI device". The authors' reply further confirms that this work still stays at the stage of material characterization. Experimentally, it is still a long way from material to a device in which the quantized Hall resistance can be observed. Therefore, I cannot support the publication in Nature Communications.

Response to Reviewer 1

Comment of the reviewer: Monolayer honeycomb lattice systems with strong spin-orbit coupling (SOC) are long-sought candidates for a stable quantum spin Hall (QSH) insulator due to their potential to host a large SOC gap. However, realizing such samples requires sophisticated monolayer fabrication techniques such as epitaxial growth and intercalation. Tilgner et al. studied a 2D bismuth layer embedded in a SiC substrate and monolayer graphene using a recently developed technique (refs. 12,13). The authors used normal incidence x-ray standing wave (NIXSW) techniques and angle-resolved photoemission spectroscopy ARPES, aided by density functional theory (DFT) calculations to characterize the structural and electronic properties. While the validity of the NIXSW data analysis is difficult to judge since the authors state that the raw data and analysis process are published elsewhere, it seems that the overall data appear to be systematically analyzed and of high quality. In particular, the NIXSW results, which provide quantitative atomic positions near the surface, are impressive. Before recommending the manuscript for publication in Nature Communications, I have several technical questions and suggestions for improvement in terms of transparency, clarity, and impact of the study.

Our response: We would like to thank the reviewer for their very positive feedback, especially highlighting the “impressive” NIXSW results. The NIXSW data set is now available at arXiv, see first point in the the following list of answers to their individual questions and suggestions.

Comment of the reviewer: 1) Given the general policy on data transparency, the minimal dataset necessary to support the conclusions (Fig. 1g and 2b) should be provided.

Our response: The reviewer is right, the NIXSW data were not available in sufficiently transparent form, but finalizing the detailed publication of the NIXSW imaging data had been delayed and was only cited as “to be published” in the original manuscript. However, now this paper, dedicated to the XSW imaging method, is available at <http://arxiv.org/abs/2504.11413>. In Sec. 2 of this new paper we describe in detail how the analysis of the NIXSW imaging data set was performed and show all the relevant data (main text Figs. 1 and 2 for the (0004) reflection, and supplement Sec. 2, Figs. S2 and S3 for all inclined reflections). In the present manuscript we have updated the corresponding citation Ref. [26].

Comment of the reviewer: 2) What do the ball sizes mean in Fig. 1g and 2b? Why does the adjacent plane appear smaller?

Our response: We are grateful for this comment, since it indicates that the presentation of the structure in this figure was not ideal. The sizes of the balls and their color saturation has no physical meaning. It simply stems from the different intensities of the maxima in the distribution maps, see, e.g., Fig. 1 (d,e), which in turn is caused by the fact that we can only consider a finite number of coefficients in the Fourier sum. Figs. 1 (g) and 2 (b) were reproduced directly from these distribution maps by cutting of all intensity below a certain threshold, and linearly scaling the remaining intensities. Then Figs. 1 (g) and 2 (b) were put together as a composite image of the corresponding density maps, in certain colors to indicate the atomic species. The scaling we applied, however, was exaggerating the (artificial) differences in the intensities of the maxima. We have improved this procedure now by using a slightly lower threshold for the cut-off, leading to more similar ball sizes, and using logarithmic scaling for the color scale, which avoids the exaggeration of the intensity differences. As a consequence of the larger ball radii, it is now clearer which atoms are located in the same plane, since atoms in the front are partly covering those in the back, rendering a pseudo-3D picture. We believe that this illustrates the real structure much better than before. In the manuscript Figs. 1 (g) and 2 (b) have been replaced and the logarithmic scaling is mentioned in the figure caption.

Comment of the reviewer: 3) Can the graphene capping layer be exfoliated? This is crucial for the potential application of the QSH device.

Our response: The reviewer is right, for employing the full potential of a QSH device the protective graphene layer is not ideal, mainly because it is metallic. Exfoliation of epitaxial graphene on SiC is possible and has been studied before (e.g., Nano Lett. 2008, 8(12), 4320; Science 2013, 342, 833; Nanoscale, 2014, 6, 889) but predominantly with a focus on achieving a high-quality

graphene sheet after exfoliation.

When putting the focus on the bismuthene layer, it will be necessary to protect it against the environment. After exfoliation, this protection would be lifted, and hence it might be more promising to modify the protection layer rather than removing it. Possible options might be the use of wide-bandgap materials like hBN instead of graphene. Also functionalizing the graphene layer in chemical reactions (Crystals 2022, 12(4), 439), or oxidizing by a plasma-treatment (Sensors 2016, 16(8), 1281) may be suitable to make the capping layer electronically inactive, but preserve its protection. However, such an experimental study is certainly beyond the scope of the present manuscript.

Comment of the reviewer: 4) The successful X-ray standing wave measurements suggest high sample uniformity, but can the authors quantify the uniformity of the bismuthene layer (e.g., uniform region size, concentration of disorder)

Our response: Based on our main experimental methods, ARPES and NIXSW, both of which have a footprint on the order of a few 100 μm , we can confidently state that the surface is homogeneous across the whole sample size. In particular, measurements were carried out on a number of different spots on the sample (sample size $5 \times 10 \text{ mm}^2$) in preparation of the NIXSW experiments. On this length scale, the sample surface is homogeneous in the sense that the ratio of bismuthene and H-intercalated QFG regions is constant all over the surface. Recent synchrotron-based ARPES measurements confirmed this down to a spatial resolution below 15 μm . The investigation of the sample uniformity on a smaller length scale is subject to ongoing investigations by low-energy electron microscopy (LEEM), which we started recently.

To address the reviewer's comment, we have added the following text on the uniformity of the sample surface in the Method section (page 8):

"Judging from the primary experimental methods employed, that are ARPES and NIXSW, we can state that the samples are uniform in the sense that the ratio of the areas covered by bismuthene and H-intercalated QFG domains is constant across the entire sample. The footprint of the incident beams of both methods is in the range of a few 100 μm ."

Comment of the reviewer: 5) The reason why the backfolded graphene replica does not appear in the precursor phase should be discussed. I would suspect that, in bismuthene, Bi is partially depleted after hydrogenation, leading to stronger interaction between graphene and the SiC surface.

Our response: It is true that the graphene replica is not visible in the ARPES data recorded for the precursor phase. This is mainly due to the surface unit cell of both phases. While both, the β and the bismuthene phases are $(\sqrt{3} \times \sqrt{3})R30^\circ$ reconstructed, the H-intercalated QFG is (1×1) reconstructed and occurs only after hydrogenation, as a parasitic phase on the bismuthene sample. The graphene replica seen in Fig. 2 d,f stems mainly from the H-intercalated QFG (1×1) phase. The $(\sqrt{3} \times \sqrt{3})R30^\circ$ reconstructed domains scatter only in a second order process to this position in reciprocal space, and hence the corresponding replica is very weak. Before hydrogenation (in Fig. 1 b) it is not visible. After hydrogenation, it contributes the lower, weaker of the two replicas, while the much stronger, upper one stems from the (1×1) reconstructed H-QFG regions.

To address this point, we added the following text to the manuscript at that location where the replicas are mentioned for the first time (page 3):

"Note that the replicas are not observed for the precursor phase (Figure 1b) since the $(\sqrt{3} \times \sqrt{3})R30^\circ$ superstructure of the β phase requires a second order process to produce scattering in this range of the k-space, making its intensity very small. In contrast, the (1×1) superstructure within the H-QFG domains yields the corresponding scattering vector in first order, leading to an enhanced intensity of the replicas related to G_H ."

Response to Reviewer 2

Comment of the reviewer: In this work, N. Tilgner et al. report on the formation of a 2D bismuthene honeycomb layer at the graphene/SiC interface. The 2D bismuthene phase is transformed from a precursor phase by hydrogenation, while the precursor phase is obtained by Bi intercalation. Combining the ARPES results and DFT calculations, the 2D bismuthene is regarded as a promising candidate for a quantum spin Hall insulator. The 2D bismuthene phase is stable in air due to the protection of the graphene sheet above and the substrate below, and it can be switched to the precursor phase by de-hydrogenation. The findings in this work are practically interesting. Before the manuscript can be considered for publication in Nature Communications, several issues require clarification.

Our response: We are very grateful for the reviewers positive feedback and address their individual comments below.

Comment of the reviewer: 1. The NIXSW imaging results show that the Bi atoms in the bismuthene precursor phase are exclusively located on the T4 sites, while in the bismuthene phase, the Bi atoms occupy the T1 sites. Why do Bi atoms move from T4 sites to T1 sites after hydrogenation?

Our response: The reviewer raises a very important question with this comment, since this move from T4 sites to T1 sites is indeed the key finding of our study. It represents the main structural change in the phase transition and enables the formation of bismuthene. Obviously, the driving force behind this move is the hydrogen, which saturates the Si dangling bonds partially: Based on our findings we conclude that hydrogen exhibits a greater affinity for the Si dangling bonds in comparison to Bi. Due to the large electron affinity of atomic hydrogen, we consider this plausible, at least as long as Bi occupies the T4 site where it saturates three Si dangling bonds. But expelling Bi from this site changes its bonding configuration. On the T1 site it saturates only one dangling bond, and additionally forms Bi-Bi bonds in a graphene-like honeycomb arrangement, which leads to an additional stabilization of this bonding configuration. Thus, hydrogen is not able to completely expel Bi from the interface, which would lead to a fully H-intercalated sample. Instead, hydrogenation and dehydrogenation allow to switch between the precursor phase and the bismuthene phase, which are presumably both metastable. Hence, the driving force behind the phase transformation is the interplay between adsorption energies of Bi and H on triply and singly coordinated adsorption sites w.r.t. the Si dangling bonds.

We have added a short paragraph addressing this point on page 4 in the manuscript:

“As our experimental results suggest, the driving force behind this change of the adsorption site is the hydrogenation. The affinity to saturate Si dangling bonds appears to be higher for H than for Bi, but not to the extent that Bi can be completely expelled from the interface. Bi is apparently repelled from its triple coordinated T_4 sites, but also stabilised at the T_1 site by the single bond to the Si surface atoms. The formation of Bi-Bi bonds in the Bi honeycomb structure of bismuthene may also contribute to this stabilisation.”

Comment of the reviewer: 2. The ARPES results in the main text exhibit the band structures of bismuthene precursor and bismuthene. And ARPES data of the Bi α phase before and after hydrogen treatment are shown in the supplementary material. For comparison, what is the band structure of graphene/SiC system before Bi intercalation?

Our response: We believe that this question is, at least partly, based on a misunderstanding. Before intercalation, the graphene layer is not (quasi) freestanding but still covalently bonded to the substrate. It is therefore referred to as zeroth-layer graphene (ZLG), and shows no distinct Dirac cones or metallic states. Its band structure has been extensively studied in the literature (e.g., Phys. Rev. B 2008, 77, 155303, Phys. Rev. Lett. 2009, 103, 246804). The ZLG does not play a particular role in the present work, but occurs only at the initial stage of the sample preparation, and in form of parasitic domains when producing the β phase (by annealing from the α phase). We believe that this aspect is sufficiently discussed in the manuscript, e.g., in the Methods section.

Comment of the reviewer: 3. What is NIXSW imaging result of the bismuthene on 6H-SiC substrate?

Our response: This is an interesting question the reviewer raises here. We do not expect significant differences in the

structural arrangements of Bi in the β and in the bismuthene phase, but rather the same outcome, namely the move from T4 to T1 sites. If this was not the case, ARPES (and also XPS) data would show clear differences for the measurements on the two substrates. But this is not the case. Except some shifts in energy due to different doping levels, the ARPES data are very similar for 4H and 6H-SiC, as to be seen in Fig. 3 of the manuscript. In particular the Bi Dirac cone is clearly visible in all cases, indicating that the bonding configuration of Bi allows the formation of bismuthene also in 6H-SiC, which is only possible with Bi on the T1 site.

However, different surface terminations of the 4H and 6H crystals might have some influence (6H-SiC exhibits two additional stable surface terminations S3 and S3*, see e.g., *Adv. Funct. Mater.* 2020, 30, 2004695), and the different unit cell size for both polytypes requires the use of a different set of inclined reflections. This certainly causes different results from NIXSW, but more on a technical rather than a physical level. For the first NIXSW imaging experiment we therefore selected the most promising and (in terms of surface terminations) most simple system, that is 4H-SiC. Another NIXSW imaging experiment on 6H-SiC might be a suitable project for the future, if synchrotron beamtime becomes available, but it must also be considered that for a sufficiently large data set at least 2-3 days of beam time are necessary, owing to the large number of inclined reflections that have to be measured. This also explains why we are not able to present an NIXSW imaging data on 6H-SiC at this stage.

Response to Reviewer 3

Comment of the reviewer: The work discusses the formation of a two-dimensional bismuthene honeycomb layer, a prospective quantum spin Hall insulator, sandwiched between a graphene layer above and the SiC substrate below. The generation of quantum spin Hall insulator is certainly an important issue for the for applications of spintronics. The authors claim that a single layer of elemental Bi, formed by intercalation of an epitaxial graphene buffer layer on SiC(0001), can be reversibly switched between an electronically inactive precursor state and a bismuthene state. In the latter, the Bi honeycomb layer is a prospective quantum spin Hall insulator. However, I have three concerns and wish they could be clarified.

Our response: We are very grateful to the reviewer for their careful and accurate review of our manuscript, and we address the three concerns in detail below.

Comment of the reviewer:

1. Validation of Topological Protection

The topologically protected properties are only indirectly verified via ARPES, lacking direct evidence from quantized conductance or non-local transport experiments. It is recommended to supplement the study with transport measurements on nanoribbon devices or theoretical calculations of edge-state bands in nanoribbons to confirm the quantum spin Hall effect.

Our response: We fully agree with the reviewer, a direct experimental proof of the quantum spin Hall effect would be required in order to confirm the topologically protected properties of the bismuthene. However, such a non-local transport measurement on nanoribbon devices is clearly far beyond the scope of the actual study. Even the preparation of protected bismuthene nanoribbons, sandwiched between graphene and SiC in a similar way as the bismuthene layer in the present manuscript, requires an enormous additional effort. Moreover, this kind of transport measurements on such a sample would certainly yield data and information that deserve one (or even several) original independent publications.

However, the second option suggested by the reviewer—theoretical calculations—has been realized. The topologically non-trivial nature of bismuthene on SiC has been demonstrated in previous theoretical works by calculating the \mathbb{Z}_2 topological invariant, which distinguishes topological ($\mathbb{Z}_2 = 1$) from trivial ($\mathbb{Z}_2 = 0$) insulators. These studies explicitly determined $\mathbb{Z}_2 = 1$, thereby confirming the quantum spin Hall phase in this system (Science 2017, 357, 287; Nano Futures 2019, 3, 045002; J. Phys. Mater. 2022, 5, 035002). Furthermore, the non-trivial topological character has also been identified through the analysis of edge states crossing the bulk band gap in finite nanoribbons of the bismuthene on SiC (New J. Phys. 2015, 17, 025005).

We have improved the manuscript in this regard, by adding a sentence in the introduction (page 1 and 2) which states this more clearly, and by adding the references [12, 13].

“For bismuthene on SiC, its topologically non-trivial nature has been established in previous theoretical works by calculating the \mathbb{Z}_2 topological invariant, which distinguishes topological ($\mathbb{Z}_2 = 1$) from trivial ($\mathbb{Z}_2 = 0$) insulators. These studies explicitly determined $\mathbb{Z}_2 = 1$, thereby confirming the quantum spin Hall phase in this system [10, 12, 13]. In addition, the topological character of the system has been further confirmed by the presence of helical edge states crossing the bulk band gap in finite nanoribbons [7].”

Comment of the reviewer:

2. Reversible Switching Stability

While the reversible switching between the precursor and bismuthene states is mentioned, the cycling stability (e.g., structural/performance repeatability after multiple hydrogenation/dehydrogenation cycles) remains unaddressed. Additional data demonstrating the robustness of switching over multiple cycles would strengthen the claims.

Our response: The reviewer raises an important point regarding the stability of the switching process. While the process of dehydrogenation is rather straightforward and fast—it can be carried out by simple annealing in UHV—the hydrogenation process is much more complex. It requires annealing of the sample in ultra-pure H_2 in a dedicated tube furnace at 880 mbar, including sample transfers etc. Therefore, carrying out a very large number of switching cycles is too time consuming. Instead, we draw

our conclusion on the reversibility of the switching from the striking similarity of the ARPES data obtained from two complete hydrogenation-dehydrogenation cycles, in which we detected no signs of sample degradation at all:

In the Figure below we compare the ARPES maps of the bismuthene phase in the first (a) and second cycle (b). In between the sample was completely dehydrogenated (as proven by Fig. S6(e) of the updated supplement), and hydrogenated again. There are no significant differences visible, in particular, the Bi Dirac bands are not degrading. Also in the Bi XPS spectra (panel (c) of the Fig. below), no changes are detectable. The same is valid when comparing the ARPES maps of the precursor state before and after the first cycle (Fig. 1b of the main paper and Fig. S6(e) of the updated supplement). Since no degradation at all can be detected after these two cycles, we are confident that the switching can be performed at least several times, as stated in our manuscript.

We adapted Sec. 5 of the supplement in order to demonstrate this more clearly. In the revised version, Fig. S6 now contains ARPES and XPS data of both the β phase and bismuthene phase from both cycles, and the discussion was extended accordingly.

Figure 1: Electronic structure of bismuthene after different hydrogenation/dehydrogenation cycles. (a) Energy-momentum cut along the $\Gamma - K_G$ direction of graphene after the first preparation. The bismuthene contribution is clearly visible around the $K_{\sqrt{3}}$ point. (b) Analogous energy-momentum cut as in (a) after dehydrogenation and rehydrogenation of the sample, where the QSHI state is switched off and on again. (c) Corresponding Bi 4f XP spectra.

Comment of the reviewer:

3. Impact of Graphene Protection on Transport

Although the bismuthene layer exhibits a wide topological gap under strong spin-orbit coupling, the metallic Dirac cones in the graphene overlayer might interfere with electronic transport. The authors should clarify whether the quantum spin Hall edge states would experience scattering into the graphene layer, potentially degrading the dissipationless spin transport properties.

Our response: The reviewer raises a severe point: does the conducting graphene layer degrade the dissipationless spin transport via bismuthene? We believe that this effect indeed occurs for any systems with an metallic protection layer. Consequently, in order to avoid this effect, it may be necessary to replace the protection layer with some wide-bandgap materials like, e.g., hBN, or by modifying it in a decent way. Oxidizing the graphene layer by plasma-treatment (Sensors 2016, 16(8), 1281) or functionalize it in chemical reactions (Crystals 2022, 12(4), 439) may be options to make the capping layer electronically inactive, but preserve the protection of the bismuthene. However, we believe that such an experimental study is beyond the scope of the present manuscript, but may be a very interesting topic for future investigations.

Response to Reviewer 4

Comment of the reviewer: The manuscript presents a study on the reversible switching of a bismuthene-based quantum spin Hall insulator intercalated between graphene and SiC(0001). While the work demonstrates competent materials engineering and thorough characterization, the core findings represent an incremental advance over prior studies (e.g., Reis Science 2017, Schmitt Nat. Commun. 2024), although the material is different. The work would be more appropriately suited for specialized materials journals, where detailed studies of material stabilization and property modulation are regularly featured. So, I do not recommend this article to be published in Nature Communications.

Our response: We appreciate the reviewer's comment, even though they do not recommend our article for publication in Nature Communications. The reviewer believes our work contains only an "incremental advance" over existing literature, and cites two preliminary papers from the same group, Reis et al., on unprotected bismuthene on SiC, and Schmitt et al, on indinene/ SiC. While both publications have high impact in the field (and are adequately cited in our manuscript), we believe that our present work also contains highly relevant new insights and represents an significant step ahead towards a QSHI device. The most significant advances of our paper are, firstly, the use of the NIXSW imaging technique yielding a precise quantitative characterization of the precursor and the bismuthene phase. In the literature there is no such direct and unambiguous way of solving the bonding configuration between the Si surface atoms and the QSHI-species, neither for Bi nor for In. This bonding configuration, however, is the key aspect for the formation of the 2D bismuthene structure. Secondly, the importance of the hydrogenation step for conversion of the precursor phase into the bismuthene phase is demonstrated by the reversible switching between the two structures, which in principle allows the QSHI state to be turned ON or OFF. To the best of our knowledge, also this feature has not been reported before. These aspects have been recognized and appreciated by the other three reviewers. We would like to invite reviewer 4 to join their colleagues in recommending the publication of our manuscript in Nature Communications.